# Pyrethroid Susceptibility in *Stomoxys calcitrans* and *Stomoxys indicus* (Diptera: Muscidae) Collected from Cattle Farms in Southern Thailand

**DOI:** 10.3390/insects13080711

**Published:** 2022-08-07

**Authors:** Sokchan Lorn, Warin Klakankhai, Pitunart Nusen, Anchana Sumarnrote, Krajana Tainchum

**Affiliations:** 1Agricultural Innovation and Management Division, Faculty of Natural Resources, Prince of Songkla University, Songkhla 90110, Thailand; 2Department of Foundation Year, University of Puthisastra, Phnom Penh 12211, Cambodia; 3Animal Production Innovation and Management Division, Faculty of Natural Resources, Prince of Songkla University, Songkhla 90110, Thailand; 4Department of Entomology, Faculty of Agriculture at Kamphaeng Saen, Kasetsart University, Kamphaeng Saen Campus, Nakhon Pathom 73140, Thailand

**Keywords:** insecticide susceptibility, pyrethroid, stable fly, *Stomoxys calcitrans*, *Stomoxys indicus*

## Abstract

**Simple Summary:**

Stable flies (*Stomoxys* spp.) are important blood-sucking insect pests worldwide that cause problems in various animal production systems. The most common species found in Thailand are *Stomoxys calcitrans* (Linnaeus, 1758) and *Stomoxys indicus* Picard, 1908 (Diptera: Muscidae). Stable flies can be controlled by many methods, with the main approaches relying on insecticides. However, insecticide resistance has been widely reported in stable flies. This study addresses related gaps in knowledge, especially regarding pyrethroid insecticides, which are used in and around agricultural farms and in households. Adult stable flies of each species were collected from cattle farms in southern Thailand, and a susceptibility test was carried out using a World Health Organization (WHO) cone bioassay. We recorded the number of knockdowns at 30 and 60 min and mortality at 12 h and 24 h after exposure. The mortality range for *S. calcitrans* and *S. indicus* was up to 95.83% and 100%, respectively. Insecticide sensitivity in the pyrethroid phenotypes was demonstrated in the population of flies from the southern provinces of Thailand. Given the stable fly populations in Thailand, not only should insecticides currently used to protect livestock be continued, but novel pest management methods should also be introduced, considering additional semiochemical toxicity to improve the effectiveness of stable fly control.

**Abstract:**

The susceptibility to six pyrethroid insecticides (permethrin, deltamethrin, alpha-cypermethrin, cypermethrin, lambda-cyhalothrin, and bifenthrin), each at the recommended concentration, was evaluated for two stable fly species—*Stomoxys calcitrans* (Linnaeus, 1758) and *Stomoxys indicus* Picard, 1908 (Diptera: Muscidae)—through tarsal contact using a World Health Organization (WHO) cone bioassay procedure. The field populations of *S. calcitrans* were collected from the Songkhla and Phattalung provinces, while *S. indicus* were collected from the Phattalung and Satun provinces in Thailand. The stable flies were exposed to insecticide-treated filter paper for 30 min, and their knockdown counts at 30 min and 60 min and mortality counts at 12 h and 24 h were recorded. The *S. calcitrans* and *S. indicus* Songkhla and Phattalung populations were moderately susceptible to pyrethroids, as indicated by the 24 h mortality. Nonetheless, the Satun population of *S. indicus* was completely susceptible to permethrin, with 100% mortality, and showed the lowest susceptibility to deltamethrin and bifenthrin. The results indicate the generally low susceptibility of stable flies to pyrethroids in the southern provinces of Thailand.

## 1. Introduction

Stable flies (Diptera: Muscidae) are common blood-sucking insect pests worldwide. There are two common species in Thailand: *Stomoxys calcitrans* (Linnaeus, 1758) and *Stomoxys indicus* (Picard, 1908) [1,2,3,4,5]. Stable flies inflict painful bites that stress livestock, causing economic losses in the form of weight loss and reduced milk production, among other losses. These insects are biological vectors and mechanical carriers of particular pathogens, which can cause problems in various animal production systems [3,6]. Moreover, stable fly bites can be a considerable annoyance to humans, especially along the beaches in West Florida, where they can severely affect the tourist industry. In the USA, economic losses caused by stable flies have been estimated at > 2000 million dollar annually [7].

Climate changes affect the life-history traits of many hematophagous insects. Stable fly activity is associated with environmental factors—both biotic and abiotic—and shows increased activity when the ambient temperature is >30 °C and the relative humidity is <50% [8]. They can be controlled by many methods, but the main methods rely on the use of insecticides. Global warming also has an impact on the insect response to insecticides [9]. However, several studies have reported that insecticides did not provide effective stable fly control [10,11]. The reduced susceptibility to insecticides in a fly population can be explained by a selection effect leading to insecticide-resistant fly strains [12]. In 1958, the insecticide resistance of stable flies to DDT (dichloro-diphenyl-trichloroethane), dieldrin, dilan, and methoxychlor has been recorded [13]. The resistance in stable flies to pyrethroid (permethrin) and organophosphate (dichlorvos and stirofos) was first reported in Kansas [12]. A few years later, stable flies from South-eastern Nebraska, NE, USA, showed low susceptibility to permethrin, stirofos, and methoxychlor [10], and a population from Florida was also reported to have low susceptibility to permethrin [11]. Then, *S. calcitrans* isolated in southwestern France was shown to be resistant to five pyrethroids (cypermethrin, deltamethrin, fenvalerate, lambda-cyhalothrin, and permethrin) [14]. Insecticide resistance detection by molecular technique has been recently reported in stable flies from the United States, Costa Rica, France, and Thailand. The positive *kdr* and *kdr*-*his* alleles of the dry stable fly samples from Nakhon Ratchasima Province, Thailand, were found to represent those populations resistant against pyrethroid [15].

As pyrethroid insecticides are typically highly toxic to insect pests but harmless to mammals, over the past decade, pyrethroids have been used in Thailand to control insect pests in both the agricultural and public health sectors. However, there has been no prior report on the biological assay for phenotypic insecticide susceptibility of stable flies in Thailand, or on the level of resistance in stable fly populations. The current study addresses these gaps in the literature, especially regarding pyrethroid insecticides that are widely used in and around agricultural farms and households.

## 2. Materials and Methods

### 2.1. Stable Fly Sampling

Adults of two common local species of stable flies in southern Thailand —namely, *Stomoxys calcitrans* and *Stomoxys indicus*—were collected. The *S. calcitrans* were collected from Songkhla (SON) and Phattalung (PHA) provinces, while *S. indicus* were collected from Phattalung (PHA) and Satun (SAT) provinces during September–November 2018, which was the wet season for the area [5] (Table 1; Figure 1). These three study sites are known for cattle farm learning centers and, on two farms that belong to the universities, insecticide treatments for pest control are applied, while one farm pursues organic farming practices. Adult stable flies of both sexes were captured, between 08:00 and 18:00, with a sweeping net and a mouth aspirator, and were kept alive and transferred to clean insect cups and later to insect cages. Morphological identification of species was individually performed under a stereomicroscope(Olympus Corporation, Tokyo, Japan), following the taxonomic keys [3,6] for sorting by species to *S. calcitrans* and *S. indicus*. The collected live specimens were kept in net cages and were provided ad libitum 100% organic honey solution and water soaked on a cotton ball for nutrition. The cages were placed in an insectary at 27 ± 2 °C and 80% RH in the Pest Management Laboratory, Faculty of Natural Resources, Prince of Songkla University, Hat Yai Campus, Songkhla Province, Thailand, for a couple of days for their blood meal digestion before the bioassay test.

### 2.2. Insecticide-Treated Paper Preparation

Insecticide-treated filter papers were prepared using acetone solutions of synthetic pyrethroid insecticides [16,17], including a technical grade of permethrin (92.15% purity), deltamethrin (99.16% purity), alpha-cypermethrin (97.34% purity), cypermethrin (93.52% purity), lambda-cyhalothrin (97.16% purity), bifenthrin (98.35% purity; Sherwood Chemicals Public Company Limited, Suan Luang, Bangkok), or commercial-grade cypermethrin (25.00% *w*/*v*; Intergrade Trading Co. Ltd., Bangkapi, Bangkok). Absolute acetone (99.99%) was used as the solvent to dissolve each insecticide, and as the negative control. Rectangular sheets of filter paper measuring 10 cm × 10 cm (Whatman^®^ No. 1) were impregnated with 1 mL of each solution, based on the recommended concentration of each technical grade insecticide—permethrin (0.5500% *w*/*v*), deltamethrin (0.0500% *w*/*v*), alpha-cypermethrin (0.0750% *w*/*v*), cypermethrin (0.2500% *w*/*v*), lambda-cyhalothrin (0.0750% *w*/*v*) and bifenthrin (0.0625% *w*/*v*), and cypermethrin (25.00% *w*/*v*)—for residual treatment of the resting-site for fly control. The treated papers were stored at 4 °C and each was used only once.

### 2.3. WHO Cone Bioassay for Residual Contact Test

The tests were performed according to Tainchum et al. [17] with slight modifications from Salem et al. [14]. Insecticide susceptibility bioassays were performed using World Health Organization (WHO) cone test kits [16]. The WHO cone was placed on a filter paper for actual treatment or control. Five stable flies (both sexes) of each species were exposed to insecticide-treated paper for 30 min, then transferred to clean plastic cups (5 individuals per cup), and provided with a cotton pad soaked with water and honey on the lid of the cup. Six replicates were carried out for each insecticide susceptibility test. The numbers of knockdown (KD) flies were recorded, considering a rapid knockdown at 30 min and a regular one at 60 min, and mortality was observed at 12 and 24 h after treatment.

### 2.4. Data Analysis

The insecticide susceptibility status of stable flies was evaluated and interpreted following the revised WHO criteria [16,18]. The observed mortality was adjusted using Abbott’s formula when the control mortality was greater than 5% and less than 20% [19]. The interpretation of percent mortality of tested stable flies at 24 h after insecticide exposure was labelled as susceptible (mortality 98–100%), incipient resistance (mortality 90–97%), or resistant (mortality below 90%) [20]. The responses of stable flies were compared between knockdown times (30 min and 60 min) and mortalities (12 h and 24 h) by calculating the differential knockdown time (DKT) = percent knockdown at 60 min − percent knockdown at 30 min and the differential mortality time (DMT) = percent mortality at 24 h − percent mortality at 12 h. A positive differential response indicates no recovery of the tested stable flies, while a negative value indicates recovery from knockdown or apparent mortality.

The mean values of percent knockdown and mortality were tested for normality using the Shapiro–Wilk test (*p* > 0.05) and were compared using an analysis of variance at alpha = 0.05. The means were separated using Tukey’s HSD if the analysis of variance was statistically significant (*p* < 0.05). Statistical analysis was performed using SPSS software for Windows (version 10; SPSS Inc.; Chicago, IL, USA).

## 3. Results

### 3.1. Insecticide Susceptibility by Population

The pyrethroid susceptibility of two stable fly species (*S. calcitrans* and *S. indicus*) was assessed by the WHO cone bioassay. The results of bioassays are shown in Table 2 and Table 3. Overall, the knockdown at 60 min after exposure was in the range of 73.33–100% across all tested populations of *S. calcitrans* and all insecticides, whereas *S. indicus* had 40–100% knockdown levels. The mortality range was from 26.67–95.83% and from 13.33–100% for *S. calcitrans* and *S. indicus*, respectively.

The *S. calcitrans* SON population exhibited a high level of knockdown with all of the insecticides (>90%) except for bifenthrin, which had an 80.83% knockdown. Complete knockdown was found in samples tested with deltamethrin and lambda-cyhalothrin. The highest mortality rate at 24 h of tested *S. calcitrans* populations to a pyrethroid was found in the SON population with bifenthrin (at 95.83%), while the mortality caused by other insecticides was below 90% (Table 2). Commercial cypermethrin provided a complete knockdown (100%) of *S. calcitrans* in the PHA population. High knockdown levels (>90%) of *S. calcitrans* in the PHA population were observed after exposure to deltamethrin, alpha-cypermethrin, and lambda-cyhalothrin. Similar to the SON strain, the highest mortality in samples of *S. calcitrans* was caused by bifenthrin (86.67% mortality; Table 2).

For *S. indicus*, the knockdown observed at 60 min after exposure to insecticide was the highest in both PHA and SAT populations with permethrin. The SAT population, when tested with alpha-cypermethrin, had a high knockdown rate (96.67%), while relatively low knockdown levels were observed with lambda-cyhalothrin in both the PHA and SAT populations. The highest mortalities of both *S. indicus* populations were determined when exposed to permethrin; namely, 96.67% and 100% for samples collected from PHA and SAT, respectively. These were considered to have incipient resistance in the PHA strain, while the SAT strain was susceptible. There was no significant difference in the mortality rate of the PHA strain when tested with permethrin and bifenthrin, but they led to higher mortalities than the other insecticides (Table 3). Mortalities caused by all other insecticides in the SAT population, except for permethrin, were not significantly different. From these results, it is obvious that all tested stable flies (strains from all sampled locations) were resistant to all pyrethroid insecticides, except for S*. indicus* from both PHA and SAT when exposed to permethrin. Additionally, *S. calcitrans* for SON population is not resistant to bifenthrin (Table 2 and Table 3).

The differential in knockdown with time (60 min and 30 min) of *S. calcitrans* were the largest in samples collected from SON and PHA when exposed to bifenthrin with 82.14% and 51.72%, respectively. The differential mortality response with time (24 h and 12 h) was the largest in samples collected from PHA tested with deltamethrin (10.34%), followed by alpha-cypermethrin and permethrin (10.00%) (Figure 2A,B). In *S. indicus*, the largest differential mortality found was in the PHA population exposed to bifenthrin (13.33%) and SAT population exposed to permethrin (10.71%). Similar to *S. calcitrans*, the differential knockdown with time was also the highest in samples of *S. indicus* from PHA and SAT with bifenthrin (63.33% and 42.85%), followed by SAT with lambda-cyhalothrin and cypermethrin (com) (−36.67%), and PHA with lambda-cyhalothrin (−34.48%) (Figure 2C,D).

### 3.2. Comparability between the Pyrethroid Insecticide Response of Two Species and Four Populations of Stable Fly

The pyrethroid insecticide response of the four populations was assessed by Tukey’s multiple range test (*p* < 0.05) (Table 4). All four populations (without considering species and province) were compared for the mortality rate (%). There was no significant difference in the mortality rate of stable flies exposed to alphacypermethrin, cypermethrin (Tec), cypermethrin (Com), and the negative control. Completely susceptible to permethrin, 100% mortality was seen in *S. indicus* SAT, which was significantly different from another three populations (73.33% mortality). Meanwhile, *S. indicus* SAT showed the significantly lowest mortality rates with deltamethrin and bifenthrin, less than half that of *S. calcitrans* SON, *S. calcitrans* PHA, and *S. indicus* PHA.

## 4. Discussion

Inappropriate and excessive use of insecticides causes contamination in the environment and induces resistance in arthropods. Their metabolites result in a reduction in contamination events and increase the challenges for those involved in livestock production, along with insecticide resistance issues [21]. Adult stable flies may be exposed to insecticides that are used during outbreak situations, such as insecticidal fogging and residual spraying to reduce the impacts of stable flies on animals. Exposure can also occur through programs for controlling other livestock ectoparasites, such as house flies (*Musca domestica* Linnaeus, 1758) (Diptera: Muscidae) [15]. The resistance of stable flies to commonly used insecticide classes has been previously reported for organochlorines, organophosphates, and pyrethroids in the U.S. and some European countries [11,12,22]. Synthetic pyrethroids have been used extensively in Thailand for the control of agricultural and livestock pests, as well as against disease vectors [23,24,25]. Resistance to pyrethroids was first reported in stable flies in Kansas cattle feedlots in 1994 [12], and the presence of resistant stable fly populations in another area was reported later on [11,14,17]. No prior study focused on the susceptibility of stable flies to insecticides has been conducted in Thailand; furthermore, no standard bioassay has yet been recommended for stable fly testing, unlike for the testing of mosquitos or house flies. In addition, this is the first report in the world to study the insecticide susceptibility status of *S. indicus*. This fly species has been found to be the second-most numerous species in all the study sites and showed crepuscular activity in a previous report [5]; therefore, the study of the susceptibility of this species to insecticides can be considered worthwhile.

This study was carried out to investigate the status of pyrethroid susceptibility in stable flies sampled from different sites in the southern part of Thailand. Our findings showed that *S. calcitrans* was the most susceptible to bifenthrin, while *S. indicus* was the most susceptible to permethrin. Bifenthrin is a non-alpha-cyano pyrethroid insecticide and acaricide recommended by the WHO for indoor residual spraying. It has a relatively low irritant and knockdown effect on mosquitoes, thus allowing the mosquitoes to rest on treated surfaces for a longer period for exposure to a lethal dose and, hence, leading to a high mortality rate [26]. These properties could have an impact on the stable fly population density if complete spraying coverage is achieved in a community. The highest permethrin susceptibility in this study was seen in *S. indicus*, and the new combination of fipronil and permethrin has been shown to provide excellent repellency and insecticidal efficacy for at least 5 weeks against *S. calcitrans* [27]. The knockdown resistance (*kdr*) associated with permethrin has been detected in stable flies from the United States, Costa Rica, France, and Thailand [15]. Regarding the susceptibility to pyrethroids, the results of this study demonstrated that the recommended concentration of any of the phenotypical tested pyrethroids (recommended concentration of commercial-grade substance) was not completely capable of controlling the two field populations of *S. calcitrans*; meanwhile, interestingly, *S. indicus* in the Satun sample was still susceptible to permethrin compared to other pyrethroids. The collection site in Satun Province was located in an organic local cow farm, which had not been treated with insecticides. However, resistance to most tested pyrethroids was observed in samples collected from this site. This may be explained by the dispersal of resistant stable flies that may occur more readily in nearby dairy farms or agricultural areas, where such selection pressure was present, impacting the effectiveness of chemical control at nearby farms. The low susceptibility of stable flies to pyrethroids seen in this study suggests that current control practices relying on only one of these insecticides alone are inadequate for further deployment. The recommended concentrations of pyrethroids for controlling stable flies should be adjusted, in order to achieve proper control and elimination. However, the phenotypic insecticide susceptibility status of stable flies in other areas of Thailand has not been reported. The susceptibility of different populations of stable flies to insecticides may be affected by various factors, such as geographic variations, breeding habitat, the health and age of stable flies, and pattern of insecticide use. Further investigations are needed to identify the susceptibility and the potential forces driving resistance in stable flies in Thailand.

The insecticide resistance of stable flies in this study may have consequences regarding the emergence of cattle and buffalo diseases in southern Thailand, especially arthropod-borne viral (arboviruses) diseases, such as lumpy skin disease (LSD), meaning that the ordinary insecticide control program is not as effective as it could be. The first case occurred on 29 March 2021, with an LSD outbreak involving beef cattle farms in Roi-Et Province, Northeastern Thailand [28]. Then, the outbreak spread all around the country [29]. The FAO team has reported that the economic impact of LSD on South, East, and Southeast countries was estimated to be up to USD 1.45 billion in direct losses of livestock and production. Vaccination could assist in the prevention of LSD, but there is still an absence of large-scale vaccination targeting susceptible livestock [30]. So, effective vector control can serve as a support to improve the reduction in disease transmission through host–vector contact.

Insecticide resistance develops with the evolutionary process of natural selection, which adapts insect populations to insecticides. The main mechanisms involve either mutation within the target site of the insecticide and/or an increased rate of insecticide detoxification [31]. To confirm insecticide resistance in a stable fly population, molecular techniques could assist in identifying the prevalence of such modifications in each population of stable flies. A recent report by Olafson et al. [15] has indicated that a low frequency of the kdr allele was observed in a stable fly population from Wang Nam Khiao, Nakhon Ratchasima, Thailand. The resistance mechanisms should be elucidated to support the estimation of resistance and to design an effective strategy for the proper control of the target pests. Moreover, standard insecticide resistance detection assays for stable flies should be developed, with improved protocols, in order to facilitate the collection of baseline information.

Further classes of insecticides should be examined and monitored to generate complementary data on the level of resistance, and possible evolution in the level of resistance in stable flies should be observed. The control of stable fly populations could be achieved by using an integrated approach or integrated vector management (IVM), involving innovations in trapping, sanitization measures, parasitoid insects, botanical insecticides, and the application of efficient chemical compounds. Furthermore, climate change has already been proven to affect the evolution of insecticide resistance in wild insects. Inducing genetic changes to the thickness or composition of the insect cuticle may affect insecticide penetration [9].

The phenotypic pyrethroid insecticide susceptibility observed in this study conclusively demonstrated resistance in most stable fly populations of the Southern provinces in Thailand. The results of this study indicate that the stable fly populations in Thailand should be further monitored, not only regarding the current insecticides used to protect livestock, but also for other factors that possibly drive the phenotypic and genetic adaptations of the population, appropriate IVM procedures, novel semiochemical toxicity, and/or different mechanisms of insecticide resistance, in order to improve the effectiveness of stable fly control. Furthermore, new standard techniques for insecticide susceptibility detection in stable fly samples should be developed, as these could be essential tools to consistently measure insecticide resistance.

## Figures and Tables

**Figure 1 insects-13-00711-f001:**
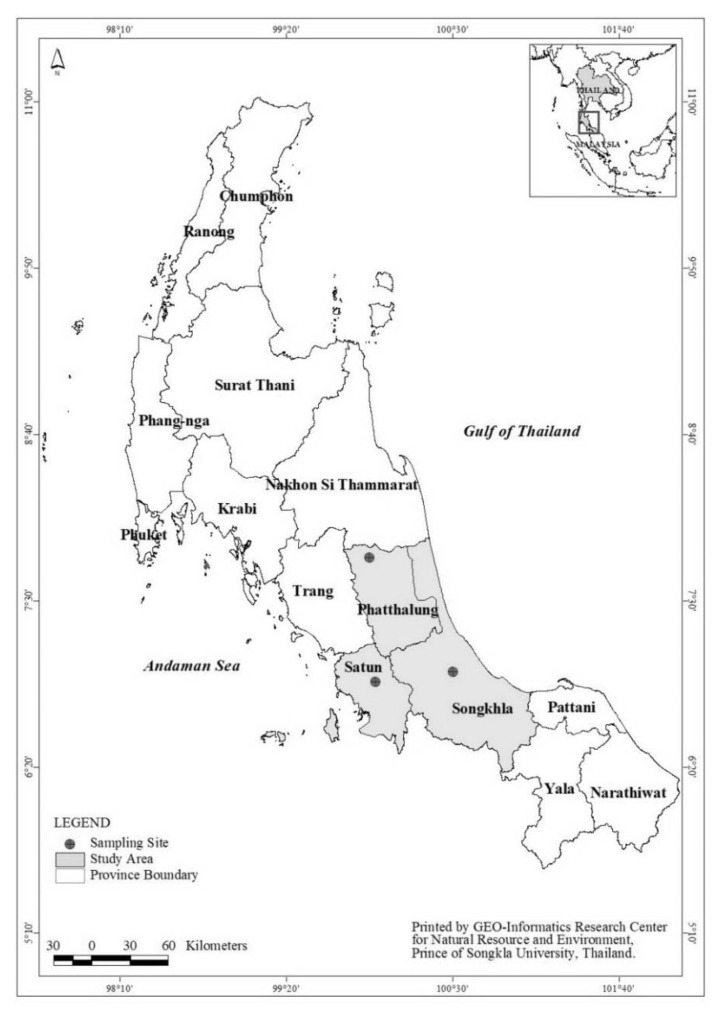
Collection sites of *Stomoxys calcitrans* and *Stomoxys indicus* located in three provinces: Phatthalung, Satun, and Songkhla, Southern Thailand.

**Figure 2 insects-13-00711-f002:**
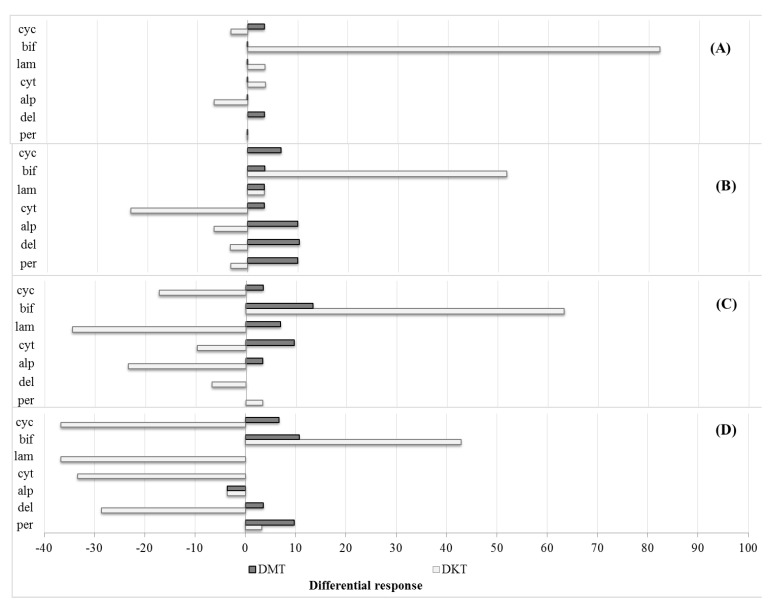
The differential response of *Stomoxya calcitrans* ((**A**) Songkhla, (**B**) Phatthalung) and *Stomoxys indicus* ((**C**) Phatthalung, (**D**) Satun) exposed to pyrethroids (per = permethrin, del = deltamethrin, alp = alphacypermethrin, cyt = cypermethrin (tec.), lam = lambdacyhalothrin, bif = bifenthrin, cyc = cypermethrin (com.)) at two knockdown observations (60 min and 30 min after exposure) and mortalities (24 h and 12 h); DMT: differential mortality time, DKT: differential knockdown time.

**Table 1 insects-13-00711-t001:** The collection sites of stable flies located in three provinces: Songkhla (SON), Phatthalung (PHA), and Satun (SAT). The associated characteristics and insecticide use are summarized.

Collection Site	Reference Point	Characteristics
Songkhla (SON)	7°00′11.6″ N	Dairy farm (near swine, poultry, and goat farms) located in Prince of Songkla University (PSU), Hat Yai District, Songkhla Province Organophosphate treatment
	100°30′15.4″ E	
Phatthalung (PHA)	7°48′13.1″ N	Dairy farm located in Thaksin University (TSU), Pa Phayom District, Phatthalung Province Pyrethroid treatment
	99°55′20.0″ E	
Satun (SAT)	6°55′49.2″ N	Organic local cow farm, located in Somyot Wechasit cattle breeding farm, Udai Charoen, Khuan Kalong District, Satun Province Untreated site
	99°57′47.5″ E	

**Table 2 insects-13-00711-t002:** Wild adult *Stomoxys calcitrans* (males and females), sampled from Songkhla (SON) and Phattalung (PHA) populations, mean knockdown (KD) and mortality rates (percentage ± SE) within 60 min and 24 h following 30 min exposure to recommended concentrations of six technical-grade and one commercial-grade insecticide in WHO cone bioassay.

Population	Insecticide ^†^	Number Exposed (*n*)	Dead	Mean Percent Responses *		Status ^‡^
				KD	Mortality	
SON	Permethrin 0.5500% (T)	31	23	96.67 ± 3.33 ^a^	73.89 ± 4.42 ^a,b^	R
	Deltamethrin 0.0500%(T)	30	14	100.00 ± 0.00 ^a^	46.67 ± 8.43 ^b,c^	R
	Alpha-cypermethrin 0.0750%(T)	30	10	90.00 ± 6.83 ^a^	33.33 ± 13.33 ^c,d^	R
	Cypermethrin 0.2500%(T)	28	10	93.33 ± 4.22 ^a^	35.56 ± 6.13 ^c,d^	R
	Lambda-cyhalothrin 0.0750%(T)	29	22	100.00 ± 0.00 ^a^	76.67 ± 9.54 ^a,b^	R
	Bifenthrin 0.0625%(T)	28	27	80.83 ± 7.57 ^a^	95.83 ± 4.17 ^a^	I
	Cypermethrin 25.00%(C)	30	13	96.67 ± 3.33 ^a^	43.33 ± 6.15 ^b,c^	R
	Control	30	1	0.00 ± 0.00 ^b^	3.33 ± 3.33 ^d^	-
PHA	Permethrin 0.5500% (T)	30	22	83.33 ± 8.03 ^a^	73.33±4.43 ^a,b^	R
	Deltamethrin 0.0500%(T)	29	20	93.33 ± 6.67 ^a^	67.50 ± 9.11 ^a,b^	R
	Alpha-cypermethrin 0.0750%(T)	30	13	93.33 ± 4.22 ^a^	43.33 ± 3.33 ^c,b^	R
	Cypermethrin 0.2500%(T)	30	8	76.67 ± 12.02 ^a^	26.67 ± 4.22 ^c,d^	R
	Lambda-cyhalothrin 0.0750%(T)	30	19	96.67 ± 3.33 ^a^	63.33 ± 8.03 ^a,b^	R
	Bifenthrin 0.0625%(T)	29	25	73.33 ± 15.20 ^a^	86.67 ± 6.67 ^a^	R
	Cypermethrin 25.00%(C)	30	13	100.00 ± 0.00 ^a^	43.33 ± 14.06 ^b,c^	R
	Control	30	0	0.00 ± 0.00 ^b^	0.00 ± 0.00 ^d^	–

* Means followed by the same letter(s) within a column are not significantly different at 5% level of significance (*p* ˂ 0.05), according to Tukey’s comparison test, ^†^ T: technical grade; C: commercial grade, ^‡^ R: resistant to insecticide; I: incipient resistance to insecticide; S: susceptible to insecticide.

**Table 3 insects-13-00711-t003:** Wild *Stomoxys indicus* (males and females) adults, from Phattalung (PHA) and Satun (SAT) populations, percent knockdown (KD) at 60 min and 24 h mortality (percentage ± SE) after exposure to an insecticide at recommended concentration for six technical-grade and one commercial-grade insecticide.

Population	Insecticide ^†^	Number Exposed (*n*)	Dead	Mean Percent Responses *		Status ^‡^
				KD	Mortality	
PHA	Permethrin 0.5500% (T)	30	29	100.00 ± 0.00 ^a^	96.67 ± 3.33 ^a^	I
	Deltamethrin 0.0500%(T)	30	8	76.67 ± 6.15 ^a,b^	26.67 ± 8.43 ^c,d^	R
	Alpha-cypermethrin 0.0750%(T)	30	8	73.33 ± 9.89 ^a,b^	26.67 ± 8.43 ^c,d^	R
	Cypermethrin 0.2500%(T)	31	17	76.19 ± 11.01 ^a,b^	52.50 ± 14.01 ^b,c^	R
	Lambda-cyhalothrin 0.0750%(T)	29	4	47.50 ± 11.95 ^b^	13.33 ± 6.67 ^c,d^	R
	Bifenthrin 0.0625%(T)	30	21	63.33 ± 9.55 ^a,b^	70.00 ± 15.28 ^a,b^	R
	Cypermethrin 25.00%(C)	29	11	83.33 ± 8.03 ^a,b^	38.33 ± 6.54 ^b,c,d^	R
	Control	29	1	0.00 ± 0.00 ^c^	3.33 ± 3.33 ^d^	–
SAT	Permethrin 0.5500% (T)	31	31	100 ± 0.00 ^a^	100 ± 0.00 ^a^	S
	Deltamethrin 0.0500%(T)	28	8	66.67 ± 13.33 ^a,b,c^	29.17 ± 8.98 ^b,c^	R
	Alpha-cypermethrin 0.0750%(T)	28	14	96.67 ± 3.33 ^a,b^	49.17 ± 10.68 ^b^	R
	Cypermethrin 0.2500%(T)	30	11	58.61 ± 9.05 ^b,c^	35.56 ± 14.05 ^b,c^	R
	Lambda-cyhalothrin 0.0750%(T)	30	12	40.00 ± 10.33 ^c,d^	40.00 ± 5.16 ^b,c^	R
	Bifenthrin 0.0625%(T)	28	15	46.67 ± 14.59 ^c^	55.00 ± 9.57 ^b^	R
	Cypermethrin 25.00%(C)	30	13	60.00 ± 7.30 ^a,b,c^	43.33 ± 9.55 ^b,c^	R
	Control	27	2	0.00 ± 0.00 ^d^	6.67 ± 4.22 ^c^	–

* Means followed by the same letter(s) within a column are not significantly different at a 5% level of significance (*p* ˂ 0.05), according to Tukey’s comparison test, ^†^ T: technical grade; C: commercial grade, ^‡^ R: resistant to insecticide; I: incipient resistance to insecticide; S: susceptible to insecticide.

**Table 4 insects-13-00711-t004:** Comparison of the mean number of pyrethroid insecticide responses (mortality rates at 24 h after insecticide exposure) of the four populations of stable flies (1 = *Stomoxys calcitrans* SON; 2 = *Stomoxys calcitrans* PHA; 3 = *Stomoxys indicus* PHA; 4 = *Stomoxys indicus* SAT).

Chemical	Test populations *				Statistic
	1	2	3	4	
Permethrin 0.5500% (T)	73.33 ^b^	73.33 ^b^	73.33 ^b^	100 ^a^	*F* = 2.196 *p* = 0.120
Deltamethrin 0.0500%(T)	46.67a ^b^	67.50 ^a^	67.50 ^a^	29.17 ^b^	*F* = 3.278 *p* = 0.042
Alpha-cypermethrin 0.0750%(T)	33.33 ^a^	43.33 ^a^	43.33 ^a^	49.17 ^a^	*F* = 0.317 *p* = 0.813
Cypermethrin 0.2500%(T)	35.55 ^a^	26.67 ^a^	26.67 ^a^	35.55 ^a^	*F* = 2.069 *p* = 0.137
Lambda-cyhalothrin 0.0750%(T)	76.67 ^a^	63.33 ^a,b^	63.33 ^a,b^	40.00 ^b^	*F* = 2.196 *p* = 0.120
Bifenthrin 0.0625%(T)	95.83 ^a^	86.67 ^a^	86.67 ^a^	55.00 ^b^	*F* = 1.230 *p* = 0.325
Cypermethrin 25.00%(C)	43.33 ^a^	43.33 ^a^	43.33 ^a^	43.33 ^a^	*F* = 23.621 *p* = 0.000
Control	3.33 ^a^	0.00 ^a^	0.00 ^a^	6.67 ^a^	*F* = 0.000 *p* = 0.000

* Means followed by the same letter(s) within a row are not significantly different at a 5% level of significance (*p* ˂ 0.05), according to Tukey’s comparison test.

## Data Availability

All datasets presented in this study are included in the article and can be provided by the authors upon reasonable request.

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
