# Peer review of "Pyrethroid Susceptibility in Stomoxys calcitrans and Stomoxys indicus (Diptera: Muscidae) Collected from Cattle Farms in Southern Thailand"

_insects, 2022, doi:10.3390/insects13080711_

Round 1

Reviewer 1 Report

This manuscript presents important results concerning the sensitivity of two Stomoxys species to different pyrethroids. This is the first work on this subject concerning the species Stomoxys indicus.

Major comments
It would be appropriate for the authors to provide some explanation of the differences in sensitivity between deltamethrin and permethrin, particularly for S. indicus from Satun province. This observation has been previously reported for some mosquito species.

Why were the two species not collected strictly from the same provinces?

According to WHO recommendations, 100 to 125 insects should be tested. Why were only 30 flies tested on average per insecticide? The weight of each insect (dead or alive) is all the more important in the evaluations of the phenotypic status and the statistical tests. This can affect the obtained results possibly.

Again, according to WHO recommendations, dilution in acetone and silicone oil is necessary for phenotypic testing. The authors should justify why they did not follow this recommendation. Furthermore, and not allowing comparison with previous work, why was the exposure only 30 min and not 1 h (WHO recommendations)?

Minor comments

The used concentrations of pyrethrinoids should be presented in mg/cm² and not in % w/v

The legend of the table 4 is incomplete

The reference 32 is not in the list of references

Author Response

We would like to express our sincere thanks to the reviewer (s) for all the valuable comments to improve this manuscript.

Reviewer 1_Comments and Suggestions for Authors

This manuscript presents important results concerning the sensitivity of two Stomoxys species to different pyrethroids. This is the first work on this subject concerning the species Stomoxys indicus.

Major comments

1.1 It would be appropriate for the authors to provide some explanation of the differences in sensitivity between deltamethrin and permethrin, particularly for S. indicus from Satun province. This observation has been previously reported for some mosquito species.

Response: Thank you for the comment. The explanation of the differences in sensitivity between deltamethrin and permethrin, particularly for S. indicus from Satun province was added in the discussion section.

1.2 Why were the two species not collected strictly from the same provinces?

Response: The two species of the stable fly were planned to collect strictly from three provinces with the same procedure as you querying. The field population abundance in each species in each study site is limited (see Lorn et al. 2020). A sufficient number for bioassay and data analysis was being considered. The trial of wild-caught stable fly mass-rearing in a laboratory has been prepared to increase the number of population (F1-F5 progeny) and to obtain a standardized sample of the wild population for use in bioassay tests. The attempt to colonize a wild-caught stable fly was unsuccessful. However, an explanation has been provided in the discussion section.

 References

Lorn S, Ratisupakorn S, Duvallet G, Chareonviriyaphap T, Tainchum K. Species composition and abundance of Stomoxys spp. (Diptera: Muscidae) in peninsular Thailand. J. Med. Entomol. 2020;57(1):252-258. doi: 10.1093/jme/tjz128. PubMed PMID: 31349364.

1.3 According to WHO recommendations, 100 to 125 insects should be tested. Why were only 30 flies tested on average per insecticide? The weight of each insect (dead or alive) is all the more important in the evaluations of the phenotypic status and the statistical tests. This can affect the obtained results possibly. Again, according to WHO recommendations, dilution in acetone and silicone oil is necessary for phenotypic testing. The authors should justify why they did not follow this recommendation. Furthermore, and not allowing comparison with previous work, why was the exposure only 30 min and not 1 h (WHO recommendations)?

Response: We greatly appreciate your concerns about the study methodology of WHO to conduct this experiment under the corresponding control conditions (location, the population of the stable flies, the concentration of insecticide…etc.). Our field study design was performed following our previous well-designed methods which were published by Salem et al. (2012) and Tainchum et al. (2018). We do agree that only 30 flies tested on average per insecticide may not access the WHO criteria.  Therefore, we believed that a sufficient number of replicates will help minimize this variation. It would be also interesting to see the real incidence from the field and this may be a research gap for future experimental studies. 

Reference

Salem A, Bouhsira E, Lienard E, Melou AB, Jacquiet P, Franc M. Susceptibility of two European strains of Stomoxys calcitrans (L.) to cypermethrin, deltamethrin, fenvalerate, lamda-cyhalothrin, permethrin and phoxim. Intern. J. Appl. Res. Vet. Med. 2012;10:249-257.

Tainchum K, Shukri S, Duvallet G, Etienne L, Jacquiet P. Phenotypic susceptibility to pyrethroids and organophosphate of wild Stomoxys calcitrans (Diptera: Muscidae) populations in southwestern France. Parasitol. Res. 2018;117:4027-4032. doi: 10.1007/s00436-018-6109-y. PubMed PMID: 30324257.

Minor comments

1.4 The used concentrations of pyrethroids should be presented in mg/cm² and not in % w/v

Response: This should be depended on the objective of the study. The concentration of insecticide in a recent study could be presented in % w/v. This concentration unit (% w/v) is shown according to the recommendation concentration and is directly associated with a mixture of insecticide and solvent (absolute alcohol).

1.5 The legend of table 4 is incomplete

Response: The title of table 4 has now been corrected and modified “Comparison of the mean number of pyrethroid insecticide responses (mortality rates at 24 h after insecticide exposure) of four populations of stable flies (1 = Stomoxys calcitrans SON; 2 = Stomoxys calcitrans PHA; 3 = Stomoxys indicus PHA; 4 = Stomoxys indicus SAT).”

1.6 The reference 32 is not in the list of references

Response: We are extremely grateful to the reviewer for pointing out the error. The citation/reference number has now been corrected and modified throughout the manuscript according to your suggestion with a total of 31 references cites.

Reviewer 2 Report

Congratulations for you research. The paper is very importante to knowledgment about insects control using chemical compouds. I will pointed out some suggestions to improve the paper.

1. I suggest a review in references. There is some mistakes in the numbers cited in the text and the author listed.

2. Table 4 has the title incomplete.

3. Tables 2 and 3 should be presented separately to a better comprehension of the results. The data should be organized by region.

Author Response

We would like to express our sincere thanks to the reviewer (s) for all the valuable comments to improve this manuscript.

Reviwer 2_Review Report Form

Congratulations for your research. The paper is very important to acknowledge insect control using chemical compounds. I will point out some suggestions to improve the paper.

2.1 I suggest a review of references. There are some mistakes in the numbers cited in the text and the author listed.

Response: The citation/reference number has now been corrected and modified throughout the manuscript according to your suggestion with a total of 31 references cites.

2.2 Table 4 has the title incomplete.

Response: The title of table 4 has now been corrected and modified “Comparison of the mean number of pyrethroid insecticide responses (mortality rates at 24 h after insecticide exposure) of four populations of stable flies (1 = Stomoxys calcitrans SON; 2 = Stomoxys calcitrans PHA; 3 = Stomoxys indicus PHA; 4 = Stomoxys indicus SAT).”

2.3 Tables 2 and 3 should be presented separately for a better comprehension of the results. The data should be organized by region.

Response: The comparison between collection sites is present in Table 4.

Reviewer 3 Report

Dear authors,

Thank you so much for your manuscript. I found it very interesting; however, I suggest major corrections to improve its understanding. Also, I would like you to emphasize in the text what it is the novelty of this study because it is not enough clear on it. You can see my comments below:

-        First time that you name a species, the authority and year should be provided, also the order and family. Especially in the case of the main objects of the study (S. calcitrans and S. indicus), for example: lines 17 (simple summary), 32-33 (abstract), and 46-47 (introduction). Also, Musca domestica in line 228.

-        I have doubts about the simple summary regarding these two sentences: line 21 (“Adult stable flies of both species were collected from three provinces of southern Thailand…”) and line 26 (“not only should the current insecticides used to protect livestock be continued…”). The first one, you mentioned that the species were collected in 3 provinces, as well as in the introduction; however, in the results, you stated that you had only 2 populations of S. calcitrans and other 2 of S. indicus, then you should better explain it in line 21 or even in the methodology (line 85: were collected from three localities in Southern Thailand (Songkhla: SON, Phattalung: PHA, and Satun: SAT provinces) because you didn’t differentiate it either, and it looks that you have 3 populations of each species. For the second sentence, you stated this in the summary but after in the discussion: “The low susceptibility of stable flies to pyrethroids seen in this study suggests that current control practices relying on these insecticides are inadequate for further deployment” (lines 264-265), your statements are contradictories also.

-        Lines 70-72: “The positive kdr and kdr-his alleles of the stable fly population from Nakhon Ratchasima province, Thailand, were found to represent those populations resistant against pyrethroid”. You already indicated that there are some reports about the resistance in Thailand but after in the next paragraph, you mentioned the following: “there has been no prior report on the susceptibility of stable flies in Thailand to insecticides, or on the level of resistance in stable fly populations” (lines 78-79). Could you better explain this in the text please? For me as a reader, it is contradictory, and it makes me wonder if this study is novel.

-        Lines 75-77: “There exists a series of registered products, from pyrethrin to synthetic pyrethroid insecticides, launched for public as well as residential uses, which may induce urban runoff and water contamination, as well as deposits in the soil”. I considered this sentence out of topic.

-        Line 84: (Lorn et al., 2020). Please follow the guidelines for the references

-        Line 121: “The tests were performed according to Tainchum et al. with slight modifications [17]” Please, place the number reference after the et al. [17].

-        Line 183: “These were considered to have incipient resistance in the PHA strain, while the SAT strain was resistant.” This sentence is talking about the mortality with permethrin, so why did you mention that the SAT strain was resistant when in the table you stablished that it is susceptible about it?

-        Tables 2 and 3: please place the statistical differences as a superscript (a, b, c, …)

-        Lines 188-190: “From these results, it is obvious that all tested stable flies (strains from all sampled locations) were resistant to all pyrethroid insecticides, except for S. indicus from both PHA and SAT when exposed to permethrin (Tables 2 and 3).” Also S. calcitrans for SON population is not resistant to Bifenthrin (you placed the letter I in the table 2)

-        Lines 191-192: “The differential mortality responses of S. calcitrans with time (24 h and 12 h) were the largest in samples collected from PHA when exposed to bifenthrin (51.72%), … (Figure 2A and 2B).” This doesn’t match with the figure, in the Figure 2A you placed 51.72% as a DKT also, not as a DMT as you wrote in the text. Please correct and unify the data.

-        Table 4: there is something missing in the caption of this table?

-        Line 237: “In addition, this is the first report to study the insecticide susceptibility status of S. indicus.” First report in Thailand or Southeast Asia or in the world? Please specify.

As I mentioned before, major revisions and clarifications should be done. If they authors correct the manuscript, it will be suitable to publish in Insect journal.

Author Response

We would like to express our sincere thanks to the reviewer (s) for all the valuable comments to improve this manuscript.

Reviewer 3_Review Report Form

Dear authors,

Thank you so much for your manuscript. I found it very interesting; however, I suggest major corrections to improve its understanding. Also, I would like you to emphasize in the text what it is the novelty of this study because it is not enough clear on it. You can see my comments below:

 3.1  First time that you name a species. Especially in the case of the main objects of the study (S. calcitrans and S. indicus), for example: lines 17 (simple summary), 32-33 (abstract), and 46-47 (introduction). Also, Musca domestica in line 228.

Response: Thank you for the comment. The authority and year, order and family of each scientific name of flies “S. calcitrans (L., 1758)”, S.indicus Picard, 1908 , Musca domestica Linnaeus, 1758 and  (Diptera: Muscidae)” were added.

3.2  I have doubts about the simple summary regarding these two sentences: line 21 (“Adult stable flies of both species were collected from three provinces of southern Thailand…”) and line 26 (“not only should the current insecticides used to protect livestock be continued…”). The first one, you mentioned that the species were collected in 3 provinces, as well as in the introduction; however, in the results, you stated that you had only 2 populations of S. calcitrans and other 2 of S. indicus, then you should better explain it in line 21 or even in the methodology (line 85: were collected from three localities in Southern Thailand (Songkhla: SON, Phattalung: PHA, and Satun: SAT provinces) because you didn’t differentiate it either, and it looks that you have 3 populations of each species.

Response: This has now been corrected and modified throughout the manuscript according to your suggestion. The text was modified accordingly;

Simple Summary: Line 21 Text was modified as “Adult stable flies of each species were collected from cattle farms in southern Thailand.

                                Line 26 The word “should” was removed.

Abstract:                 Line 34 Text was modified as” Field populations of S. calcitrans were collected from Songkhla and Phattalung provinces, while S. indicus were collected from Phattalung and Satun provinces in Thailand.

Materials and Methods: Line 85 Text was modified as “The S. calcitrans were collected from Songkhla (SON) and Phattalung (PHA) provinces, while S. indicus were collected from Phattalung (PHA) and Satun (SAT) provinces.

3.3 For the second sentence, you stated this in the summary but after in the discussion: “The low susceptibility of stable flies to pyrethroids seen in this study suggests that current control practices relying on these insecticides are inadequate for further deployment” (lines 264-265), your statements are contradictories also.

Response: This has now been corrected and modified. The text in simple summary and lines 264-265 were corrected “The low susceptibility of stable flies to pyrethroids seen in this study suggests that current control practices relying on only one of these insecticides alone are inadequate for further deployment

3.4 Introduction Lines 70-72: “The positive kdr and kdr-his alleles of the stable fly population from Nakhon Ratchasima province, Thailand, were found to represent those populations resistant against pyrethroid”. You already indicated that there are some reports about the resistance in Thailand but after in the next paragraph, you mentioned the following: “there has been no prior report on the susceptibility of stable flies in Thailand to insecticides, or on the level of resistance in stable fly populations” (lines 78-79). Could you better explain this in the text please? For me as a reader, it is contradictory, and it makes me wonder if this study is novel

Response: Thank you for this question. We are missing the information which should be added to the text. In that study in Nakhon Ratchasima, the dry stable fly samples were obtained and detection of the kdr and kdr-his alleles was done in the laboratory without a biological test as the current study.

Lines 70-72: The text was corrected as” Insecticide resistance detection by molecular technique has been recently reported in stable flies from the United States, Costa Rica, France, and Thailand. The positive kdr and kdr-his alleles of the dry stable fly sample from Nakhon Ratchasima Province, Thailand, were found to represent those populations resistant against pyrethroid.

Lines 78-79: The text was corrected as “However, there has been no prior report on the biological assay for phenotypic insecticide susceptibility of stable flies in Thailand, or on the level of resistance in stable fly populations.

3.5  Introduction Lines 75-77: “There exists a series of registered products, from pyrethrin to synthetic pyrethroid insecticides, launched for public as well as residential uses, which may induce urban runoff and water contamination, as well as deposits in the soil”. I considered this sentence out of topic.

Response: This sentence has now been removed according to your suggestion.

3.6 Materials and Methods Line 84: (Lorn et al., 2020). Please follow the guidelines for the references

Response: “(Lorn et al., 2020)” have been removed.

3.7  Materials and Methods Line 121: “The tests were performed according to Tainchum et al. with slight modifications [17]” Please, place the number reference after the et al. [17].

Response: This has now been corrected and modified.

3.8 Results Line 183: “These were considered to have incipient resistance in the PHA strain, while the SAT strain was resistant.” This sentence is talking about the mortality with permethrin, so why did you mention that the SAT strain was resistant when in the table you established that it is susceptible about it?

Response: We are extremely grateful to the reviewer for pointing out the error. The word “resistance” has been replaced by “susceptible”.

3.9 Results Tables 2 and 3: please place the statistical differences as a superscript (a, b, c, …)

Response: This has now been corrected and modified.

3.10 Results Lines 188-190: “From these results, it is obvious that all tested stable flies (strains from all sampled locations) were resistant to all pyrethroid insecticides, except for S. indicus from both PHA and SAT when exposed to permethrin (Tables 2 and 3).” Also S. calcitrans for SON population is not resistant to Bifenthrin (you placed the letter I in the table 2)

Response: This has now been corrected and modified. The text “Also S. calcitrans for SON population is not resistant to Bifenthrin” was added.

3.11 Lines 191-192: “The differential mortality responses of S. calcitrans with time (24 h and 12 h) were the largest in samples collected from PHA when exposed to bifenthrin (51.72%), … (Figure 2A and 2B).” This doesn’t match with the figure, in the Figure 2A you placed 51.72% as a DKT also, not as a DMT as you wrote in the text. Please correct and unify the data.

Response: This has now been corrected and modified.

The differential in knockdown with time (60 min and 30 min) of S. calcitrans were the largest in samples collected from SON and PHA when exposed to bifenthrin with 82.14% and 51.72%, respectively. The differential mortality response with time (24 h and 12 h) was the largest in samples collected from PHA tested with deltamethrin (10.34%), followed by alpha-cypermethrin and permethrin (10.00%) (Figure 2A and 2B). In S. indicus, the largest differential mortality found was in the PHA population exposed to bifenthrin (13.33%) and SAT population exposed to permethrin (10.71%). Similar to S. calcitrans, the differential knockdown with time was also the highest in samples of S. indicus from PHA and SAT with bifenthrin (63.33% and 42.85%), followed by SAT with lambda-cyhalothrin and cypermethrin (com) (-36.67%), and PHA with lambda-cyhalothrin (−34.48%) (Figure 2C and 2D).

3.12 Result Table 4: there is something missing in the caption of this table?

Response: The title of table 4 has now been corrected and modified “Comparison of the mean number of pyrethroid insecticide responses (mortality rates at 24 h after insecticide exposure) of four populations of stable flies (1 = Stomoxys calcitrans SON; 2 = Stomoxys calcitrans PHA; 3 = Stomoxys indicus PHA; 4 = Stomoxys indicus SAT).”

3.13 Discussion Line 237: “In addition, this is the first report to study the insecticide susceptibility status of S. indicus.” First report in Thailand or Southeast Asia or in the world? Please specify.

Response: We have now modified from Thailand to “in the world”.

3.14 As I mentioned before, major revisions and clarifications should be done. If they authors correct the manuscript, it will be suitable to publish in Insect journal.

Response: We would like to express our sincere thanks to the reviewer for patiently preparing the valuable comments to improve this manuscript.

Round 2

Reviewer 1 Report

The anwers to previous comments are clear and justified.

Author Response

The answers to previous comments are clear and justified.

Response: We would like to express our sincere thanks to the reviewer for patiently preparing the valuable comments to improve this manuscript.

Reviewer 3 Report

Dear authors,

Thank you so much for the changes, the manuscript was well improved; however, still some minor corrections should be done:

-          Tables 2 and 3: please place the statistical differences as a superscript (a, b, c, …). You corrected all letters in the tables, except for S. indicus SAT population, please correct the superscripts also (Table 3).

-          Line 274-281: I consider this additional paragraph out of topic. Why do you talk now about mosquitoes when you are not naming them in the whole manuscript? If your focus is on Stomoxys, I recommend you to delete this or to better explain the relationship, also you need some references to support your new statement.

-          I recommend using a professional English proofreader service for better understand and to correct all the mistakes in the text.

Author Response

Comments and Suggestions for Authors

Dear authors,

Thank you so much for the changes, the manuscript was well improved; however, still some minor corrections should be done:

3.1   Tables 2 and 3: Please place the statistical differences as a superscript (a, b, c, …). You corrected all letters in the tables, except for S. indicus SAT population, please correct the superscripts also (Table 3).

    Response: We are extremely grateful to the reviewer for pointing out the error. This has now been corrected and modified.

3.2 Line 274-281: I consider this additional paragraph out of topic. Why do you talk now about mosquitoes when you are not naming them in the whole manuscript? If your focus is on Stomoxys, I recommend you to delete this or to better explain the relationship, also you need some references to support your new statement.

    Response: This has now been removed according to your suggestion.

3.3  I recommend using a professional English proofreader service for better understand and to correct all the mistakes in the text.

    Response: This manuscript has been successfully edited for English language and spelling by MDPI. We are grateful to the reviewer for their valuable time and constructive comments on the manuscript.

This manuscript is a resubmission of an earlier submission. The following is a list of the peer review reports and author responses from that submission.